# The Food Safety Knowledge, Attitude and Practice of Malaysian Food Truck Vendors during the COVID-19 Pandemic

**DOI:** 10.3390/healthcare10060998

**Published:** 2022-05-27

**Authors:** Wan Nor Fatihah Wan Nawawi, Vimala Ramoo, Mei Chan Chong, Noor Hanita Zaini, Ping Lei Chui, Zamzaliza Abdul Mulud

**Affiliations:** 1Department of Nursing Science, Faculty of Medicine, Universiti Malaya, Kuala Lumpur 50603, Malaysia; fatihahnawawi@um.edu.my (W.N.F.W.N.); mcchong@um.edu.my (M.C.C.); noorhanita@um.edu.my (N.H.Z.); chuipinglei@um.edu.my (P.L.C.); 2Centre of Nursing, Faculty of Health Science, Universiti Teknologi MARA, Shah Alam 42300, Malaysia; zamzaliza@uitm.edu.my

**Keywords:** knowledge, attitude, practice, food safety, food truck vendors

## Abstract

Foodborne diseases are one of the greatest public health threats, but they can be prevented by maintaining food safety practices. Although the food safety literature has been awash with studies from various food operations, there is very limited information on food safety in the food truck business. Therefore, this study aims to assess the level of knowledge, attitude, and practices related to food safety among food truck vendors. A cross-sectional study was conducted among 396 Malaysian food truck vendors using convenience sampling. Data was collected during the Recovery Movement Control Order due to the COVID-19 pandemic through a validated self-administered questionnaire and analyzed using SPSS version 25. The findings indicated that food truck vendors generally have fair knowledge (*M* = 78.8, *SD* = 9.09), a positive attitude (*M* = 94.8, *SD* = 5.95), and good practices (*M* = 84.7, *SD* = 6.62) regarding food safety during the COVID-19 pandemic. Hierarchical regression analysis further showed that food truck vendors’ level of education and knowledge of food safety are significant predictors of their food safety practices. This study provides an initial understanding of the food safety knowledge, attitude, and practices among food truck vendors and yields important information in promoting the food safety culture.

## 1. Introduction

With an estimated 420,000 deaths per year from foodborne diseases (FBD), food safety is recognized as a top public health concern worldwide, with 42% of cases reported in the Southeast Asia region alone [1]. In Malaysia, the Ministry of Health (MOH) reported an increase of 23.69% of FBD cases in 2018 compared to the previous year [2]. However, the report did not specify any food businesses involved in the FBD outbreak [3]; therefore, the extent of cases stemming solely from the food truck business remains unknown in Malaysia.

Furthermore, cases in Malaysia are often underestimated as most FBD cases go unreported, primarily because victims often do not seek appropriate medical treatment in a health facility [4]. The lack of accurate data on the occurrence of FBD makes it difficult for policymakers to improve current regulations [5]. Moreover, FBD not only affects one’s health outcomes but even a single outbreak event results in economic losses due to changes in consumer purchasing behavior and national expenditure for medical treatment [6]. Therefore, food safety practices are essential to reducing the prevalence of FBD.

Most reported FBD outbreaks in Malaysia have been from brick-and-mortar food premises [7]. In contrast, despite the thriving of the food truck industry, limited research has been conducted on food safety among food truck vendors (FTVs). To date, only one study has been conducted among food handlers in the food truck business in Malaysia [8]. However, the study only focused on hand hygiene practices, not overall food safety components. Additionally, the findings of the study cannot be considered an actual representation of the FTV population in Malaysia due to its low response rate [8]. This limited literature on food safety in the food truck industry and the emerging need for food safety during the current COVID-19 pandemic urges the need to assess food handlers’ food safety practices.

The COVID-19 pandemic, dubbed the novel coronavirus, became a global emergency in December 2019 [9] due to the human-to-human transmission that infected millions of citizens, including Malaysians. Although the initial SARS-CoV-2 outbreak at the seafood market in Wuhan, Hubei, China is suspected to have originated from the consumption of animal-derived food [10], to date, there is no evidence that COVID-19 spread via food [11]. However, there is a possibility of virus transmission through food handlers as the virus could survive for several days on the surfaces of utensils or food handling materials [11]. Proper hand-washing techniques; cleaning of raw materials, kitchen equipment, and environment; and food storage are required to combat FBD and also the possible transmission of COVID-19 virus [12].

As the COVID-19 caused a devastating pandemic with serious consequences for global health and the economy, the Movement Control Order (MCO) was announced by the Malaysian government on 18 March 2020 [13], to control the spread and transmission of the virus. Meanwhile, on 4 May 2020, the National Security Council (NSC) announced the reopening of economic sectors including the food sector under the Recovery Movement Control Order (RMCO) [14], with applications of the standard operating procedures (SOP). The SOPs for the food sector was enforced for all premises situated in a building, roadside stalls, and farmers’ markets. This includes wholesale markets, mobile food trucks, grocery stores, and hawker stores aside from restaurants.

The SOPs for the food handlers include the use of biodegradable food containers and detergents containing sodium hypochlorite. In addition, the SOPs for the food truck business include the strict operating hours (from 5.00 p.m. to 12.00 a.m.) [14]. They were also required to record employees’ body temperature upon arrival, ensure a 1-meter distance between customers, limit the number of customers at a time, and make the use of alcohol-based hand sanitizers compulsory. These SOPs have been created to ensure the safety of food and the public from the human-to-human transmission of COVID-19.

Recognizing the importance of measuring the FTVs’ awareness and behavior in relation to food safety in accordance with the new normal practices imposed by the current pandemic and to fill the gaps in the food safety literature, this study aimed to assess the level of knowledge, attitude, and practices (henceforth referred to as KAP) of FTVs regarding food safety and the impact of their knowledge and attitudes on their food safety practices during the COVID-19 pandemic. The results of this study are expected to serve as a reference for the food truck industry and relevant bodies to transform food safety approaches to evolve the industry towards the new normal practices.

## 2. Materials and Methods

### 2.1. Data Collection

This study employed a quantitative, cross-sectional design. It is noted that the number of vendors registered with the Ministry of Food Safety and Quality of Malaysia [15] for the food truck business from 2013 to 2018 was 2001 vendors. Using this estimate, the sample size for the current study was determined using the Daniels (1999) protocol with a 95% confidence level and a 5% margin error. A total of 396 samples were determined after accounting for the 20% attrition rate. Although the food trucks initially operated at different times in different areas, as this business has increased, local authorities have designed a number of areas or parks for the FTVs to operate and where the food trucks can congregate. Data was collected from the designated areas around the capital of Malaysia using a convenient sampling method. As a multicultural country, the Malaysian food truck industry offers a wide range of food and beverage varieties. All FTVs who met the inclusion criteria (aged 18+, understand English or Malay) were invited to participate in this study during the RMCO due to the COVID-19 pandemic from September 2020 to March 2021.

The FTVs were approached prior to their business operations or in their free time while waiting for the customer. The purpose of this study was explained and copies of the questionnaire with cover letters were distributed after obtaining their consent. The FTVs were assured of the confidentiality of their responses and the voluntary nature of their participation. The FTVs were given sufficient time (approx. 30 min) to answer the questionnaire. Data collection was performed according to the SOPs enforced during the RMCO period such as the application of face masks, social distancing, hand sanitation, and use of individual pens to answer the questionnaire. The reporting of this study is in accordance with the Statement on Strengthening the Reporting of Observational Studies in Epidemiology (STROBE).

### 2.2. Instruments

This study used a self-administered questionnaire for data collection. The questionnaire consists of scales adapted with permission from previous studies by Sani and Siow [16] and Lee et al. [17]. Minor changes were made to sentence structures and items to meet the need of the current study. Whereby relevant items from the SOPs imposed on the food sectors in response to the COVID-19 pandemic were integrated into the knowledge and practice sections of the questionnaire. Thus, the final questionnaire consisted of four main sections: sociodemographic characteristics (10 items), food safety knowledge (41 items), attitudes (10 items), and practices (30 items) during COVID-19 pandemic. The questionnaire was translated from English to Malay by linguistic experts for cross-cultural use according to the guidelines of the World Health Organization (WHO) [18] and then validated by 10 experts in the fields of nursing, food microbiology, food safety, and public health.

The knowledge component has four sub-domains: personal hygiene [19], foodborne diseases [20], food storage and handling [21], and cleaning and sanitizing [22]. The response options were ‘True’, ‘False’, or ‘Don’t know’.

The attitude scale includes 10 items to assess FTVs’ belief in their role in handling food during the COVID-19 pandemic. A 5-point Likert scale from 1 (strongly disagree) to 5 (strongly agree) was used. The scores were summed to represent the total score for the attitude scale. The following are examples of items included in the scale: “Producing safe food is more important to me than tasty food”; “Safe food handling during pandemic is an important part of my job responsibilities”; and “I am willing to change my food handling behaviors when I know they are incorrect”.

Lastly, the food safety practice was assessed under four main domains: personal hygiene (8 items), foodborne disease preventive measures (6 items), food storage and handling (9 items), and cleaning and sanitation techniques (7 items). The response options were ‘Yes’, ‘No’ or ‘Not sure’.

The reliability of the instruments was tested with a pilot study of 42 FTVs, the results were: knowledge (Kuder Richardson Formula 20 = 0.74), attitude (Cronbach’s alpha = 0.92), and practice (Kuder Richardson Formula 20 = 0.81); indicating all scales had acceptable internal consistency.

### 2.3. Ethical Consideration

This study was conducted in accordance with the Declaration of Helsinki and the Caldicott Principle. Meanwhile, ethical approval was obtained from the Medical Research Ethics Committee (MREC) of University Malaya Medical Centre, Kuala Lumpur, Malaysia (MRECID.NO: 2020217-8289) and permission was granted by the local authorities before the data collection procedure.

In addition, all FTVs received an information sheet about the study and were asked to provide written informed consent, with assurances that their anonymity and confidentiality would be maintained throughout the study. Finally, the safety of the soft and hard copy data was ensured.

### 2.4. Data Analysis

The collected data were carefully sorted and screened before being entered into the Statistical Package for the Social Sciences (SPSS) version 25 for further analysis. The KAP scores were normally distributed; accordingly, an independent *t*-test, one-way ANOVA, Pearson’s correlation, and hierarchical regression tests were performed in the study. The descriptive results of the constructs were presented as mean and standard deviation (*SD*), while a significance level of 5% (*p* < 0.05) was assumed to assess the relationships of the constructs.

All negative statements in the questionnaire were reversed-coded before the scoring process to obtain coherent data analysis and accurate data representation. Responses for the knowledge and practice scales were dichotomously recoded as follows: “1 (Do not know)” and “2 (No)” were combined into “0 = Incorrect”, whereas “3 (Yes)” was scored as “1 = Correct”. The five-point Likert scale responses for the attitude scale were computed to obtain the total score.

The scores for each section were summed and the level of food safety KAP was presented in percentage form, wherein a higher percentage reflected a better result. The total scores were interpreted based on the Bloom’s cut-off points [23]: good knowledge/positive attitude/good practice (above 80.00%), fair knowledge/neutral attitude/moderate practice (60.00–79.99%), and poor knowledge/negative attitude/poor practice (below 60.00%).

## 3. Results

A total of 396 FTVs participated in this study, yielding a 100% response rate. The respondents were predominantly Malaysians (85.4%), male (65.4%), single (70.5%), and had at least secondary education (68.7%). The average age of the FTVs was 25.58 (*SD* = 6.31 years), ranging from 18 to 62 years. In terms of work experience, more than one-third of the FTVs had more than 2 years of experience in the food industry (38.4%); however, they had less than 1 year of experience in the food truck business, with an average of 1.68 (*SD* = 1.57 years). Furthermore, almost three-quarters of the FTVs had received food safety training (71.2%). Most of them had also been vaccinated against typhoid disease (79.3%) and have regular medical check-ups (60.4%).

### 3.1. Level of KAP of Food Safety among FTVs during COVID-19 Pandemic

This study found that FTVs generally had a fair level of knowledge (*M* = 78.84, *SD* = 9.09, range = 36.59 to 95.10), a positive attitude (*M* = 94.84, *SD* = 5.95, range = 60 to 100), and good practices (*M* = 84.65, *SD* = 6.62, range = 56.67 to 96.67) towards food safety during the COVID-19 pandemic (Table 1). 

A detailed analysis of the FTVs’ knowledge of food safety during the COVID-19 pandemic revealed the highest score in the cleaning and sanitation sub-domain (*M* = 91.14, *SD* = 8.68). Based on the item analysis, all FTVs appeared to be aware of the importance of cleaning storage areas, covering dustbins, and separating food and cleaning products to reduce the risk of food contamination. In contrast, the lowest score was found for food storage and handling, with a mean of 70.18 (*SD* = 11.39).

Interestingly, almost all FTVs (98.5%) knew that social distancing of at least 1 m must be observed in business operations to break the chain of COVID-19 infection. The majority of FTVs (90.4%) had knowledge that washing hands with soap and water, or using hand sanitizer could prevent the spread of viruses. A large number of FTVs (86.6%) were aware that the use of personal protective equipment (mask, gloves, etc.) is very important to reduce the risk of infectious diseases, including COVID-19.

With regard to attitudes towards food safety, a similar mean value of 4.75 was found for most items. This study found that most FTVs agreed that managing food safety is their important responsibility. More than two-thirds of FTVs indicated that they would explicitly agree to increase their knowledge of food safety and were willing to change incorrect food-handling behaviors, ensuring safe food is more important than tasty food. However, 1.3% of FTVs indicated that they were unwilling to attend food hygiene/safety training/courses, while 13.1% were uncertain *(M* = 4.65, *SD* = 0.75). The majority of FTVs strongly agreed that their knowledge of food safety is very important.

Additionally, item analysis for the food safety practices revealed the highest score for personal hygiene (*M* = 93.71, *SD* = 10.14). Meanwhile, the lowest scoring was noted for items in the FBD subdomain, contributing to a lower mean score (*M* = 65.97, *SD* = 15.68), with most FTVs answering the question “Do you work when you are sick and having symptoms of infectious disease?” with a yes.

With regards to the COVID-19 pandemic, a vast majority of FTVs (99.5%) reported they wash their hands hygienically before preparing food, after touching raw ingredients, and after using the toilet/restroom to reduce cross-contamination of food. Most FTVs (98.5%) reported washing and sanitizing the food preparation area and cooking equipment with detergents containing sodium hypochlorite, as recommended by the MOH. Lastly, 94.2% of the FTVs had reported wearing a mask at all times during food truck business operations during the COVID-19 pandemic.

### 3.2. KAP of Food Safety among FTVs during COVID-19 Pandemic According to Sociodemographic Characteristics

Table 2 displays the food safety KAP scores during the COVID-19 pandemic according to sociodemographic characteristics of the FTVs. Knowledge scores of food safety were found to differ significantly according to the FTVs’ nationality, food safety training, and regular medical check-ups. Specifically, the independent *t*-test noted significantly higher knowledge scores among Malaysian FTVs (*t* = 5.788, *d* = 0.58, *p* < 0.001), had undergone food safety training (*t* = −4.730, *d* = 0.75 *p* < 0.001), and regularly attend medical check-ups (*t* = −4.054, *d* = 0.52, *p* < 0.001).

Besides, FTVs’ work experience in the food industry and their food safety training showed statistically significant differences in attitude scores, whereby the ANOVA test revealed that FTVs with 1–2years of experience in the food industry have higher attitude scores, *F* (2, 250) = 4.527, *ηp*^2^ = 0.03, and *p* = 0.002, while trained FTVs had a significantly positive attitude towards food safety during the COVID-19 pandemic. (*t* = −3.794, *d* = 0.61, *p* < 0.001).

Lastly, there were statistically significant differences in the food safety practices during the COVID-19 pandemic between the FTVs’ nationality, education level, food safety training, and medical check-ups. Significantly higher scores for the practices were observed among Malaysian FTVs (*t* = 3.640, *d* = 0.88, *p* = 0.001), FTVs who had attended college/university (*F* = 13.249, *ηp*^2^ = 0.05, *p* < 0.001), FTVs with food safety training (*t* = −2.709, *d* = 0.41, *p* = 0.007), and FTVs who had regular medical check-ups (*t* = −5.306, *d* = 0.73, *p* < 0.001).

### 3.3. The Relationships between the KAP of Food Safety among FTVs during COVID-19 Pandemic

Pearson’s correlation test found statistically significant weak positive relationships between knowledge and attitude (*r* = 0.139), knowledge and practices (*r* = 0.233), and attitude and practices (*r* = 0.104) pertaining to FTVs’ food safety during the COVID-19 pandemic (all *p* < 0.05).

A further analysis using a hierarchical regression test was performed to determine the impact of knowledge and attitude on food safety practices among FTVs during the COVID-19 pandemic (Table 3). In addition, the analysis identified education level and food safety training as covariates at an alpha level of 0.25, while a food safety practice score was set as the dependent variable. FTVs’ education level and food safety training were entered at step one (Model 1) to control for demographic variable responses. Then, knowledge was entered in step two (Model 2), followed by their attitudes towards food in step three (Model 3).

Results revealed that the final model was significant (*p* < 0.001), with an adjusted explanatory power, *R*^2^, of 0.092. The FTVs’ characteristics in Model 1 accounted for 6.8% of the variation in the food safety practice, while knowledge of food safety accounted for an additional 3.2% when included in the second model (*R*^2^ = 0.100, *F* (4, 391) = 431.271, *p* <0.001; adjusted *R*^2^ = 0.091). Attitude’s contribution in Model 3 was not significant (*R*^2^ = 0.104, *F* (5, 390) = 358.954, *p* < 0.001; adjusted *R*^2^ = 0.092). In the final model, FTVs’ education level (*β* = 0.282, *p* < 0.001) and knowledge of food safety (*β* = 0.181, *p* < 0.001) were identified as significant predictors of their food safety practices during the COVID-19 pandemic.

## 4. Discussion

FBDs are largely preventable by maintaining the hygienic-sanitary quality of food [24]; however, adequate knowledge and attitude are deemed necessary to facilitate ideal food safety practices [25]. Grounded in this assumption, this study was conducted to primarily assess the relationship between food safety KAP in FTVs.

Consistent with the existing literature, this study found that Malaysian FTVs generally have adequate knowledge ranging from good to fair level [26,27] and good practices [16,17] regarding food safety. It was also found that KAP for food safety differed significantly based on the FTVs’ food safety training and their level of education. Previous scholars [28,29] have pointed out similar findings, believing that effective training can improve food handlers’ awareness and knowledge of food safety compliance [30]. Moreover, food handlers who participated in food safety training are more likely to exhibit a positive attitude, thereby improving food safety practices [22].

The hand hygiene practice was determined to be a food safety challenge for FTVs, as food truck equipment might not be ideal for personal hygiene accommodations [31]. However, it is enlightening that almost all the FTVs in this study reported good personal hygiene knowledge; as a result, it is relatively certain that they would practice adequate handwashing with soap and water before, during, and after food handling, or after visiting the restroom. This finding is in line with previous studies on FTVs in the United States [31] and Brazil [22]. Additionally, FTVs’ high level of hand hygiene compliance may be due to the awareness of the SOPs set by the NSC in response to the COVID-19 pandemic.

Regarding FBD, it is surprising to note that 89% of the FTVs claimed they would still work despite being sick or having symptoms of an infectious disease. This practice is highly inappropriate according to the Codex Alimentarius Commission [32], because sick food handlers may inadvertently transfer germs to prepared food or anything else they come into contact with in the food preparation area. In addition, individuals with symptoms of illness should not appear in public, especially when dealing with customers during the critical time of the COVID-19 pandemic. Previous studies have shown that the possible reasons for this are staff shortages and food handlers’ beliefs that their symptoms are not severe and that they can still get their work done [33]. Moreover, as the COVID-19 pandemic caused a burden on the FTVs who struggled with great losses in business income during those months of non-operation due to the MCO [34], the FTVs had no other choice but to continue operating even when they were not in their best health condition. Thus, the authorities need to increase awareness and monitoring to ensure compliance towards food and public safety among the FTVs.

The significant positive relationships between the KAP domains in this study are consistent with the previous literature [16,22]. These findings suggest that food handlers’ level of knowledge will influence their attitudes and practices towards safe food handling, especially during the COVID-19 pandemic. In contrast to the KAP model, however, further analysis with the regression model could not establish attitude as a significant predictor for food safety practice [25]. Nonetheless, a previous KAP food safety integrative review found that 20% of the selected studies concluded attitude was not translated into practice [35], consistent with this study. This suggests that further research is needed on the role of attitudes in the KAP model in relation to food safety. Perhaps a broader scale is needed to measure the attitudinal component.

Next, this research revealed that the level of education and food safety knowledge of FTVs are significant predictors of their food safety practices, consistent with a study in South America [36]. The latter emphasized that improving food safety practices in the food service industry is imperative but can only be achieved if food handlers acquire knowledge of food safety through education and training. Food safety training seems to be an effective tool for improving knowledge and is fundamental to proper practices. Training courses are important because they provide a guideline on the basic food handling techniques for food handlers and increase the prevention of food safety risks. Previous studies have discussed that training should be conducted at least every 6 months to a year [32] to ensure the effectiveness and compliance of food safety among FTVs.

Finally, the results of this study revealed that almost all FTVs agreed to comply with the SOPs enforced on the food sectors. This may be due to the heightened public awareness regarding the COVID-19 pandemic. There was a lot of useful information that could be obtained through various communication channels such as infographics, charts, illustrations, advertisements, and photographs, as well as the internet and social media networks related to COVID-19 and the enforcement of SOPs. In addition, this level of compliance could be hampered by fear of being fined for violating the SOPs of RMCO. This is very important because the SOPs were actually in line with the existing food safety guideline to prevent any infectious diseases.

## 5. Strengths and Limitations

This study is the first to specifically investigate the current food safety knowledge, attitudes, and practices of FTVs in Malaysia, while also examining the compliance levels towards the SOPs of the food sector during the COVID-19 pandemic. Significant relationships between knowledge, attitudes, and practices of food safety were found in this study, indicating that FTVs need to enhance their knowledge and performance of correct food safety practices. It is recommended for the food safety authorities to revise this course and establish an updated and standardized food safety guideline. Furthermore, any identified deficiency in the knowledge, attitudes, and practices of food safety in this study should serve as an initial diagnosis to guide the educational strategies aimed at the promotion of a food-safe culture in Malaysia’s food truck industry.

Although most of the FTVs in this study were formally trained, the legal requirements in Malaysia must be revised, as some of the FTVs had never attended the food safety training course organized by the Ministry of Health. Therefore, the authorities should provide an opportunity for all food handlers to participate in food safety training and ensure that they are vaccinated accordingly before allowing them to start any food-related business operations. Continuous monitoring by the authorities is recommended to ensure food safety compliance among the FTVs.

Besides that, the findings of this study are able to serve as baseline data for the healthcare sector, such as the MOH’s Food Safety and Quality Division (FSQD) and nursing. In the public health sector, nurses play an important role in advocating for correct health procedures for the community, including food safety and foodborne disease preventive measures for food handlers and consumers, therefore, the data of this study could be used in support of evidence-based nursing practices. The public health nurses could expand their role in educating the public and vendors through offline or online campaigns regarding the importance of food safety. Other than that, nurses should be directly involved in and actively contribute to food handling training organized by the MOH.

Furthermore, the findings of this study offer avenues for future research, whereby more studies are needed to assess the knowledge, attitudes, and practices of food safety among FTVs from other city councils because of their differences in license or business permit requirements, food safety facilities, and sociodemographic characteristics. Additionally, the tool used in this study can also be used by future studies for validation, so that a validated tool will be made available to researchers assessing the knowledge, attitudes, and practices of food safety among FTVs. Action research and qualitative studies could provide more in-depth findings on actual practice and rationales behind the FTVs’ attitude, which was not captured in this study.

Although this study provides several implications for the food truck business and the food safety literature, it is subject to several limitations due to its cross-sectional nature. The data collected was self-reported, which may imply response bias; therefore, future research should consider longitudinal observational studies on socio-psychological factors and environmental risk factors that may contribute to food safety practice among FTVs. In addition, the instrument-measured attitude had limited items and therefore may have influenced the results of the KAP model in this study, thus requiring further exploration of the attitude component. Finally, the data presented in this study may not be representative of FTVs in Malaysia or other regions due to the convenience sampling method and the limited geographic area from which the sample was drawn. Future research should include FTVs in other states to improve generalization and to comprehend regional differences between cultures and belief systems.

## 6. Conclusions

Foodborne diseases have been and continue to be a threat to humans in general. It is also a huge economic burden on the catering and healthcare industries. It is important that authorities and food safety officials act swiftly to improve this issue. Therefore, knowledge, attitudes, and practices of food safety among FTVs need to be addressed because they play an essential role in maintaining food safety and providing safe food to consumers. In this study, FTVs generally have fair knowledge, a positive attitude, and good practices related to food safety during the COVID-19 pandemic. Although attitude was found to have a weak positive relationship with food safety practices, the regression model suggests that attitude does not have a significant impact on the FTVs food safety practices. However, the cultivation of positive culture among the FTVs is necessary for the food safety industry. Finally, it is important to note that food safety needs to be consistent with the SOPs enforced during the pandemic, particularly in relation to personal hygiene to keep the public safe. Therefore, FTVs should be adequately trained and supervised to increase their knowledge and awareness of food safety and improve safe food practices.

## Figures and Tables

**Table 1 healthcare-10-00998-t001:** Level of KAP of food safety among FTVs during COVID-19 pandemic (N = 396).

Variables	Range of Score	Frequency (n)	Percentage (%)	Mean (*SD*)
Knowledge				78.84% (9.09)
Good	≥80.00%	193	48.7	
Fair	60.00–79.99%	187	47.3	
Poor	<60.00%	16	4.0	
Attitude				94.84% (5.95)
Positive	≥80.00%	330	83.3	
Neutral	60.00–79.99%	34	8.6	
Negative	<60.00%	32	8.1	
Practices				84.65% (6.62)
Good	≥80.00%	351	88.6	
Moderate	60.00–79.99%	42	10.6	
Poor	<60.00%	3	0.8	

**Table 2 healthcare-10-00998-t002:** KAP of food safety during COVID-19 pandemic according to FTVs’ sociodemographic characteristics (N = 396).

Characteristics	Frequency (n)	Percentage (%)	Knowledge				Attitude				Practice			
			Mean	*SD*	*t*/*F*	*p*	Mean	*SD*	*t*/*F*	*p*	Mean	*SD*	*t*/*F*	*p*
Gender					−0.690 ^a^	0.490			0.451 ^a^	0.652			0.746 ^c^	0.457
Male	259	65.4	78.61	9.075			95.04	5.848			84.85	5.863		
Female	137	34.6	79.28	9.136			94.46	6.164			84.28	7.859		
Age (in years)					0.996 ^b^	0.370			0.532 ^b^	0.588			0.252 ^b^	0.777
<23	160	40.4	79.51	8.677			94.00	5.864			84.67	5.758		
23–29	146	36.9	78.05	9.374			94.18	6.341			84.41	6.947		
≥30	90	22.7	78.94	9.339			95.82	5.470			85.04	7.499		
Nationality					5.788 ^a^	0.001 **			1.512 ^c^	0.135			3.640 ^c^	0.001 *
Malaysian	338	85.4	79.89	8.282			95.28	5.699			85.27	6.078		
Non – Malaysian	58	14.6	72.71	11.055			92.28	7.185			81.09	8.363		
Marital status					0.830 ^a^	0.407			0.167 ^a^	0.867			1.316 ^c^	0.190
Single/Divorced/Widowed	279	70.5	79.09	8.918			94.90	5.968			84.98	5.75		
Married	117	29.5	78.26	9.504			94.68	5.941			83.87	8.30		
Education level					2.966 ^d^	0.056			0.671 ^b^	0.512			13.249 ^d^	0.001 **
No formal education/Primary	46	11.6	74.44	13.533			96.34	4.932			81.67	9.047		
Secondary	272	68.7	79.30	8.493			94.38	6.226			84.45	6.462		
College/University	78	19.7	79.77	7.099			95.48	5.528			87.14	4.255		
Experience in food industry					0.986 ^b^	0.374			4.527 ^d^	0.012 *			0.631 ^b^	0.532
<1 year	102	25.8	77.91	9.797			95.84	5.172			85.26	6.591		
1–2 years	142	35.9	79.56	8.184			96.64	5.161			84.58	6.068		
>2 years	152	38.4	78.80	9.399			92.48	6.899			84.32	7.124		
Experience in food truck business					1.987 ^b^	0.139			0.181 ^b^	0.834			0.475 ^d^	0.623
<1 year	152	38.4	77.82	9.533			94.80	5.947			84.78	6.608		
1–2 years	148	37.4	79.91	8.621			95.22	5.994			84.93	5.641		
>2 years	96	24.2	78.81	8.983			94.30	5.951			84.03	7.931		
Food safety training					−4.730 ^c^	0.001 **			−3.794 ^c^	0.001 **			−2.709 ^c^	0.007 *
Yes	282	71.2	80.39	7.696			96.52	4.949			85.27	6.137		
No	114	28.8	75.01	11.144			90.70	7.545			83.13	7.490		
Regular medical check-up					−4.054 ^c^	0.001 **			−1.816 ^c^	0.071			−5.306 ^c^	0.001 **
Yes	239	60.4	80.44	7.208			95.76	5.358			86.21	4.313		
No	157	39.6	76.42	10.961			93.44	6.717			82.29	8.556		
Typhim IV vaccination					−0.922 ^a^	0.357			0.930 ^a^	0.353			−1.723 ^a^	0.086
Yes	314	79.3	79.06	8.864			94.56	6.149			84.95	6.522		
No	82	20.7	78.02	9.924			95.92	5.131			83.54	6.896		

Note: Significant, * *p* < 0.05, ** *p* < 0.001, ^a^ Independent *t*-test, ^b^ One-way ANOVA test; ^c^ Welch *t*-test; ^d^ Welch ANOVA test.

**Table 3 healthcare-10-00998-t003:** Hierarchical multiple regression predicting the practices of food safety during the COVID-19 pandemic (N = 396).

Variables	Practices of Food Safety					
	Model 1		Model 2		Model 3	
	*B*	*β*	*B*	*β*	*B*	*β*
Constant	**78.607 ****		**69.632 ****		**66.783 ****	
Education level						
Secondary	**2.696 ***	0.189	**2.082 ***	0.146	**2.183 ***	0.153
College/University	**5.206 ****	0.313	**4.597 ****	0.277	**4.679 ****	0.282
Food safety training	**1.852 ***	0.127	1.140	0.078	0.951	0.065
Knowledge			**0.** **136 ****	0.187	**0.132 ****	0.181
Attitude					0.073	0.066
*R* ^2^	0.068		0.100		0.104	
*F*	**9.555 ****		**10.829 ****		**9.030 ****	
Δ*R*^2^	0.061		0.091		0.092	

Note: N = 396; *B* = regression coefficient; *β* = beta coefficient; Δ*R*^2^ = adjusted *R*^2^; ** *p* < 0.001; * *p* < 0.05. Predictors: education level, food safety training, knowledge (continuous variable), and attitude (continuous variable). Dependent variable: Practices of food safety. Education level was represented as three dummy variables with No formal education/Primary as the reference group.

## Data Availability

Data available on request from the authors.

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
