# Peer review of "The Food Safety Knowledge, Attitude and Practice of Malaysian Food Truck Vendors during the COVID-19 Pandemic"

_healthcare, 2022, doi:10.3390/healthcare10060998_

Round 1

Reviewer 1 Report

Dear Authors,

The manuscript (healthcare-1700047) submitted for review is very interesting. I recommend it for publication after the Authors' answers to the questions and after minor corrections.

Apart from the title, I did not find any mention of the Covid-19 pandemic time and the food trucks operating during this time.

Introduction: If this article is one of the first about food trucks vendors in Malaysia, it is a pity that it is not known what kind of food is sold in this type of establishment.

Results
p.3.3. (lines 196-209): The information in this section on the 3 models identified requires discussion and explanation, not just show.

Discussion: It is a pity that the authors did not discuss the study results concerning the hygiene recommendations during the Covid-19 pandemic.

Limitation: I propose to separate the limitations and strengths of these results. Now the authors have added limitations to the conclusions section. This is not a good idea.

Conclusion: In this section, the authors should respond to the research objective set out in lines 75-79. These conclusions need to be rewritten as they are too general.

References: References are cited following the journal's rules. There is a lot of literature on this topic also in MDPI.

Despite my comments, I am pleased to recommend this manuscript for publication. I believe it addresses an important area of research in an international context.

Reviewer

Author Response

Dear reviewer,

Thank you.

Reviewer 2 Report

The research aims to assess the level of knowledge, attitude and practices related to food safety among food truck suppliers. The tool used is a self-administered questionnaire. The study is based on an excellent experimental design. Likewise, it appears to rely on solid statistical processing. However, the text does not sufficiently clarify some important methodological aspects. I refer to the ways in which the subjects included in the sample were identified, the description of the questionnaire and the methodology with which it was administered. Due to this lack of transparency, the judgment on the paper can only be suspended.

I ask the authors to complete materials and methods by reporting:

  • how the food truck vendors were identified;
  • how it was possible to reach them;
  • the questions in the questionnaire;
  • the methodology applied to administer it.

Parts of text to be corrected

24: … of food truck vendors …

Is it a repeat?

29: With an estimated 420,000 deaths …

I propose to add "per year"

84-89: As the frequency of food truck activity in Malaysia is not standardized or known in advance, all FTVs who met the inclusion criteria (aged 18 years or older, understand English or Malay language, and are available in Kuala Lumpur) were invited to participate in this study using convenience sampling during the RMCO due to the COVID-19 pandemic, from September 2020 until March 2021.

It is not clear how the FTVs who participated in the survay were identified and reached. I ask that it be specified better.

92: 2.2 Instruments

The information on the questionnaire concerns many aspects relevant to the research, nevertheless it is insufficient.

A real representation of the questionnaire is needed, in which the questions are clearly indicated. Furthermore, it is unclear how it was administered. For example: email, social media, other.

Author Response

Dear reviewer, 

Thank you very much.

Round 2

Reviewer 2 Report

The authors made substantial improvements in the description of the context, hypothesis and experimental design, as well as materials and methods, results and discussion. Regarding the improvement requests made by me previously, it is now clear how the questionnaire was administered. The description of the questionnaire used has also improved significantly, however the questions with which the attitude of the interviewees is assessed is still insufficiently described to understand and contextualize the results.

In particular, while the information collected with the questionnaire to evaluate the knowledge and the food safety practice is sufficiently clear to understand and contextualize the results, the same cannot be said for the attitude.

I therefore invite the authors to improve this point of the paper as well as they have already done for all the other aspects, so that all the scientific value that the research certainly has can be recognized.

Parts of text to be corrected

128-130: Attitudes towards food safety were assessed based on vendors’ feelings and insights into issues that could influence their compliance.

The information on the questions of the questionnaire aimed at investigating the aptitude is insufficient to contextualize the results obtained.

198: Table 1

It seems that it is necessary to standardize the form of writing fonts (≥80,00%)

267: (β=0.0.282, …

a correction is needed

306-311: In contrast to the KAP model, however, further analysis with the regression model failed to establish attitude as a significant predictor for food safety practice [25]. Nonetheless, a previous integrative review on food safety KAP noted that 20% of the selected studies concluded attitude was not translated 309 into practice [35], which is consistent with this study. This suggests that further exploration is needed on the role of attitudes in the KAP model in relation to food safety.

Attitude is a dimension characterized by the data by which it is measured. The information on the questions in the questionnaire aimed at investigating this aspect is insufficient to contextualize it.

Author Response

Dear reviewer,

Thank you.
